# Conservation Biology and Reproduction in a Time of Developmental Plasticity

**DOI:** 10.3390/biom12091297

**Published:** 2022-09-14

**Authors:** William V. Holt, Pierre Comizzoli

**Affiliations:** 1Department of Oncology & Metabolism, The Medical School Beech Hill Road, Sheffield S10 2RX, UK; 2Smithsonian’s National Zoo and Conservation Biology Institute, Washington, DC 20008, USA

**Keywords:** animal conservation, reproduction, conservation breeding, epigenetics, developmental plasticity, assisted reproductive technologies

## Abstract

The objective of this review is to ask whether, and how, principles in conservation biology may need to be revisited in light of new knowledge about the power of epigenetics to alter developmental pathways. Importantly, conservation breeding programmes, used widely by zoological parks and aquariums, may appear in some cases to reduce fitness by decreasing animals’ abilities to cope when confronted with the ‘wild side’ of their natural habitats. Would less comfortable captive conditions lead to the selection of individuals that, despite being adapted to life in a captive environment, be better able to thrive if relocated to a more natural environment? While threatened populations may benefit from advanced reproductive technologies, these may actually induce undesirable epigenetic changes. Thus, there may be inherent risks to the health and welfare of offspring (as is suspected in humans). Advanced breeding technologies, especially those that aim to regenerate the rarest species using stem cell reprogramming and artificial gametes, may also lead to unwanted epigenetic modifications. Current knowledge is still incomplete, and therefore ethical decisions about novel breeding methods remain controversial and difficult to resolve.

## 1. Introduction

The last two decades have seen a revolution in the way that scientists understand how organismal development takes place, and a realization that the DNA sequence itself represents only one part of the process. Expressing the genomic DNA sequence must be controlled by an exquisitely refined and complex series of interactions that are collectively known as epigenetic mechanisms. These typically involve important enzyme-mediated chemical modifications, such as methylation and acetylation, not only of the DNA itself and various proteins with which the DNA interacts, but also of a series of regulatory and non-coding RNA species [1]. These emerging discoveries have revolutionized our understanding of the various ways in which finely controlled gene expression influences not only offspring development, but the inheritance of some newly acquired phenotypic characteristics, much as hypothesized by Lamarck in the 19th century [2,3]. The fact that multiple phenotypes may be generated from a single genotype is often known as ‘developmental plasticity’, and typically occurs in response to external environmental cues experienced during the more plastic early stages of development [4].

Here, we aim to ask whether, and how, some of these novel discoveries might affect the ways in which some of the principles that underpin conservation biology may need to be reviewed or refocused in the light of epigenetics.

## 2. Conservation and Adaptations

The general aim of conservation biology is to inhibit population declines, minimize inbreeding as far as possible, and maintain good population health. Therefore, attempting to support reproductive success and future population fitness is an important component of this aim. However, given that the threatened species live within diverse habitats, it follows that they will experience, and respond to, the demands of the habitats in question (Figure 1).

Habitats vary enormously, and highly localized forms of conservation practice have developed. These include large-scale projects such as the establishment of marine conservation areas or large national parks, as well as some very small and local projects focused on a particular species. Captive breeding is also a well-established and widely used conservation method, being mainly employed by zoos, aquariums, and some wildlife reserves. An estimate published in 2011 suggested that one-seventh of all threatened species are held, and bred, in zoos and aquariums [5]. In fact, some small populations owe their continued existence exclusively to captive breeding, and a relatively small number have even been successfully reintroduced to their original habitats. A classic example of a successfully reintroduced captive-bred species is the Père David’s deer (*Elaphurus davidianus*) [6], which were returned to China in the 1980s, having been bred in western zoos for nearly a century. Given the situation and the diversity of species in question, the concept that species can adapt and respond successfully to all manner of environmental changes is almost incredible. It is certainly not our intention to suggest that captive-bred mammals are necessarily unsuitable for reintroduction.

Whether the long-term survival of zoo-bred and reintroduced species is advantaged or disadvantaged by developmental plasticity is, at present, an unanswerable question. While reintroduction programs for captive-bred mammals are always managed with great care, allowing animals to become acclimated to their new environments, their long-term success is likely not only to be species-dependent, but also only recognizable several generations later. At present there are few, if any, long-term studies that can shed light on this question, but the well-established black-footed ferret (*Mustela nigripes*) captive breeding and assisted reproduction program [7,8] may, in the future, provide some answers. Genetic studies have shown that artificial insemination with cryopreserved black-footed ferret semen, collected and stored several years prior to use, successfully reduced the level of inbreeding in the captive population. However, it is not yet known whether and how the levels of inbreeding may influence the eventual breeding success and survival of the wild populations.

Some caution is also relevant in this context. A recent meta-analysis of birth-origin effects on the outcomes of captive breeding across a wide range of species, including invertebrates, fish, birds, and mammals [9], showed that wild-born animals in captivity have a massive advantage (about 74%) in terms of reproductive success compared to their captive-born counterparts. These results were based on several reproductive traits, including reproductive yield (e.g., litter size), birth weight, offspring survival, and reproductive phenology (e.g., interbirth interval). Interestingly, when data were examined in relation to the captive environments, only aquaculture showed a large statistically significant mean advantage for wild-born over captive-born animals. As aquaculture tends to be focused on improving reproductive outputs, this effect was regarded as unexpected, and the authors suggested that supplementation with wild stocks would be required in order to sustain population fitness and welfare. However, other evidence obtained from aquaculture suggested that the plasticity effects involved in captive breeding could be induced so rapidly that they might actually cause maladaptation [10].

## 3. Contemporary Evolution and Developmental Plasticity

The evolutionary literature is replete with examples of species where both males and females have evolved and adapted their physiological, behavioural, and morphological characteristics, optimizing their survival, reproductive success and fitness over centuries and millennia. However, while many of these long-term adaptations have been effected via gene mutation and various kinds of evolutionary selection, many such adaptations are induced by changes in the diverse ways in which gene expression is controlled. Such changes, caused via developmental plasticity, need not be solely determined by the precise nature of the DNA sequences involved. In fact, by changing the way that the DNA sequences are controlled, species are able to respond quickly to environmental changes. For example, the complex social organization of eusocial insect societies (notably, bees, wasps and ants) is dependent on developmental plasticity, whereby genetically identical individuals develop differentially into functionally different castes [11]. This level of sophistication involves the control of gene expression during development, via DNA methylation, histone modifications and other epigenetic mechanisms [12,13] and is a beautiful example of developmental plasticity. “Polyphenism” is similar in principle, but the term is used to describe the remarkable ability of many insects and some fishes, such as the blue headed wrasse (*Thalassoma bifasciatum*) [14,15], to alter their own phenotype in response to changing environmental conditions (for review, see [16]). Locusts are notable in this context, as they can change their own behaviour and morphology, and those of their offspring, in response to cues such as crowding, which involves visual, olfactory and tactile stimuli. 

### Plasticity under Extreme Conditions

Developmental plasticity allows species to respond in the face of challenging conditions, and some have exploited that facility in ways that allow them to live successfully under the most extreme environments. Some of the developmental strategies are truly remarkable and possibly unexpected. For example, some fish species, especially the African and South American killifishes, which live in so-called “ephemeral” bodies of water that are prone to both flooding and droughts, survive successfully by enabling their embryos to undergo a series of physiological mechanisms minimizing water loss and entering a period of dormancy or diapause [17]. This strategy is so extreme that in some cases none of the adult fish remain alive during the drought, and the entire population is represented by dormant embryos. Embryos are frequently buried in mud to avoid predation, and when the drought finishes, the embryos hatch, grow rapidly and reproduce prior to the next dry period [18,19,20,21]. 

Some species that can survive in extreme habitats, such as hyper-arid deserts and rocks, oceanic deeps, salt lakes, the north and south poles, volcanoes, high mountains, and upper atmosphere, provide interesting examples of the resilience that can be attained via evolutionary plasticity. It is not surprising that some researchers are currently exploiting biological plasticity for the development of novel technologies and applications [22]. One well-known and notable example is the widely used enzyme for DNA amplification, Taq DNA polymerase. This enzyme, which is derived from the thermophilic eubacterial microorganism *Thermus aquaticus*, is unusual because of its extreme heat resistance. With a half-life of 40 min at 95 °C, it can be used in the polymerase chain reaction technique, where other less stable enzymes would be rapidly denatured.

## 4. Developmental Plasticity Is a Continuing Process

A contemporary example of developmental plasticity involves the food fish steelhead trout (*Oncorhynchus mykiss*) [23,24]. Comparison of wild and hatchery-raised trout found evidence that adaptation to a life in freshwater, rather than the ocean, could occur rather quickly. The authors identified three chromosomal regions that were associated with rapid genetic and functional adaptation to environmental change. The environmental change in question relates to historical relocation of these particular fish from native rivers in California, where a period of developmental time is normally spent in the ocean, but where, following geographical translocation to a freshwater habitat in Lake Michigan, the fishes treat the freshwater lake as a surrogate ocean. One chromosomal region was found to contain functional changes to ceramide kinase, which may have altered metabolic and wound-healing rates in the Lake Michigan steelhead. The second and third chromosomal regions encoded carbonic anhydrases and a solute carrier protein, both of which are critical for osmoregulation. Importantly, these authors found that despite their high reproductive success in captivity, the hatchery-bred fishes exhibited low reproductive success and high mortality in the wild. The authors suggested several potential reasons for this difference, including changes in egg size, growth rate, and ability to avoid predators, all of which represent selection pressures that govern the ability of populations to withstand environmental stresses and thrive into the future. A separate study [25] of the role of phenotypic plasticity in the adaptation of wild Trinidadian guppies (*Poecilia reticulata*) to a completely new environment, found that dramatically improved reproductive rates could be detected within 20–30 generations and supported a previous suggestion that phenotypic plasticity could be regarded as “contemporary evolution” [26]. 

## 5. Reproductive Forecasting and Development

Reproductive and developmental success both involve some degree of forecasting in the face of changing future environments. This helps the organism to match the path of embryonic development against expectations [27], and thus optimize the survival of subsequent generations. In fact, this relationship is at the heart of reproductive seasonality, where species have developed physiological strategies that typically synchronize breeding and parturition with climatic conditions for the optimal survival of their offspring [28]. If the forecast turns out to be inaccurate because the environment changes and defies expectations, the mismatch can have unfortunate and lifelong health consequences. 

Several amphibian species that are vulnerable to drought exploit forecasting and developmental plasticity to protect themselves in case their aquatic habitats are likely to dry out. Some may be able to speed up the rate of larval development if they detect increased salinity [29,30,31,32], while others pay attention to their choice of nest site [33]. The exact responses appear to be species specific, and not always successful if features of their environment, such as rainfall, are unpredictable [34]. 

Organisms are now known to use such developmental forecasting to predict how to prepare for possible and stressful changes in nutrition, temperature, salinity, and even the presence of environmental toxicants. Such relationships are now well recognized in humans and other mammals [35], owing mainly to a series of studies linking early life nutrition to the onset of diseases in adulthood (see, for example, [36,37]). The realization that there are systematic relationships between pre- and/or peri-conception nutritional conditions and chronic disease conditions in adults has stimulated a great deal of research in humans and other species. It has even been shown, in humans, that transgenerational effects on body condition in males are related to the father’s and grandfather’s smoking habits [38]. Similarly, experimental studies in rats have demonstrated the occurrence of transgenerational impacts of experimentally administered agricultural chemicals [39,40]. Transgenerational reproductive forecasting, sometimes known as “anticipatory transgenerational phenotypic plasticity” was also observed in relation to rates of flea infestation in a desert rodent (*Meriones crassus*) [41]. Maternal reproductive success was best when mothers and grandmothers experienced similar risks of parasitism. When both were either infested or non-parasitized, the pups would reach sexual maturity more quickly than those pups whose mothers’ infestation status did not match that of their grandmothers. 

## 6. Wildlife Conservation, Captive Breeding, and Mammalian Sex Ratios

Vertebrate populations have evolved variously sophisticated methods of adjusting the sex ratios of their offspring in response to current environmental conditions, or in anticipation of future conditions. Achieving these outcomes at the population level can only occur if individuals are able detect and respond to environmental cues experienced very early in the reproductive process, even before conception in the case of species with internal fertilization. Although many mammalian species possess the ability to skew offspring sex ratios [42], the biological mechanisms seem, at first sight, to defy statistical logic. Spermatogenesis in mammals results in two distinct, and equally sized, populations of spermatozoa, with each cell either carrying an X or a Y chromosome. Once fertilization takes place and embryogenesis begins, the presence of the Y-chromosome within the zygote activates the molecular processes involved in sex determination, and a male embryo develops. Statistically, a 50:50 ratio of male to female embryos would normally be expected, unless the females’ physiology can be influenced to interfere with sperm transport, impose sperm selection prior to fertilization, or even selectively prevent embryonic development. 

Captive-bred pigmy hippopotamus (*Choeropsis liberiensis*) present an extreme example of sex ratio skewing, with highly female-biased sex ratios at birth described in two different studies (41% and 43% males) [43,44]. This outcome appears to support the hypothesis proposed by Trivers and Willard in the 1970s [45], that as maternal condition declines, adult females tend to produce more female than male offspring. Conversely, a significantly male-biased sex ratio of Asian elephants born in European zoos between 1962 and 2006 (ratio: 0.61, *p* = 0.044) was reported in 2009 [46], as well as an even more striking male bias in elephant births following artificial insemination (0.83, *p* = 0.003). It has clearly been very difficult for researchers to find a unifying theory that explains the differing reports of biased mammalian sex ratios at birth in diverse wild and captive species (see, for example [47,48,49,50,51,52,53]), in relation to environmental conditions. However, that such mechanisms exist and may sometimes, but not necessarily always, be adaptive, is now beyond dispute. 

The mechanisms involved in sex ratio biasing are likely to involve sperm selection within the female reproductive tract, thus biasing the genetic properties of the spermatozoa that eventually fertilize the oocyte(s). Cameron et al. [54] and others [55,56,57] have critically analysed possible stages in the mammalian reproductive process that could lead to skewed sex ratios, with a focus on mechanisms that might affect sperm transport and fertilization. They concluded that the female reproductive tract employs many effective sperm-selection mechanisms that are capable of preventing, or enabling, the passage of specific sperm subpopulations. Cervical and oviductal mucus can act as physical barriers to sperm progression, and some anti-microbial defensins appear to be influential in both inhibiting or assisting sperm transport [58]. Recent studies of mouse sperm progression in vivo by direct examination of whole reproductive tracts [59] have demonstrated clearly that the spermatozoa travel as clusters, rather than as individuals, a cooperative behaviour that is also observed in other mammalian species [60]. Spermatozoa are highly responsive to a range of molecules, some as simple as the bicarbonate ion, which activates cyclic adenosine monophosphate (cAMP), induces protein phosphorylation and stimulates both capacitation and rapid flagellar movement [61,62]. Sperm entry into the oviductal isthmus from the utero-tubal junction is not only dependent on progressive motility, but is critically dependent on the exact molecular nature of the proteins expressed at the sperm surface, especially the A-disintegrin and metalloprotease group, widely known as ADAM proteins [63,64,65]. Moreover, the epithelial cell products typically prevent sperm motility and capacitation until they register that ovulation is under way [66], whereupon motility and directionality are influenced by chemotaxis [67,68] and thermotaxis [69]. Furthermore, components of oviductal fluid cause significant hardening of the zona pellucida [70,71,72], the proteinaceous coat the surrounds the oocyte, and reduce polyspermic fertilization. Seminal plasma is also known for its ability to influence the responses that the female reproductive tract to the arrival of spermatozoa [73], including the induction of de novo gene expression. Experimental ablation of the seminal vesicles in mice resulted not only in the impairment of pregnancy, but also abnormal placental function in late pregnancy, and deleterious impacts on the health of male offspring after birth [74]. 

### Oviductal Cells Can Discriminate between X- and Y-Chromosome-Bearing Spermatozoa

Those spermatozoa that successfully enter the oviductal environment are known to bind and interact with epithelial cells, which respond by stimulating de novo gene transcription [75,76,77]. Some of the newly synthesized proteins, such as heat shock proteins, support prolonged sperm viability [78], while others are more directly involved in fertilization itself. Although these effects may not necessarily result in biased sex ratios, recent evidence [79] has shown that the oviductal cells themselves are capable of differentially responding to the presence of X- and Y-chromosome-bearing spermatozoa. The authors used laparoscopic surgery to isolate the oviducts of female pigs that were in heat. Sex-sorted [80] X- and Y-bearing spermatozoa (3 × 10^5^ sperm in 100 µL volumes) were separately inseminated into the left and right oviducts, and the oviductal tissues were subsequently recovered after 24 h for genomic analysis (Figure 2). 

The outcome indicated that the oviducts possess a recognition system that distinguishes the chromosomal nature of spermatozoa (out of 501 transcripts, 54.1% were downregulated by the presence of Y-bearing spermatozoa, and 45.9% were upregulated); these data indicate that the oviducts could therefore affect gender selection prior to fertilization. The biological systems most affected by these changes were (in descending order) signal transduction, the immune system, the digestive system, and the endocrine system. 

A recent review [81] presented a summary table derived from six independent studies, showing proteins that have been reported as differentially expressed in X- and Y-bearing spermatozoa (31 proteins were upregulated in X-spermatozoa, while only 10 were upregulated in Y-spermatozoa). The authors were somewhat reticent in their interpretation of these outcomes, and concluded that further research would be required before they could be entirely convinced of the veracity of the data. 

## 7. Impacts of Gamete Cryopreservation on Offspring Development and Survival

At present, it is probably too soon to evaluate the conservation relevance of epigenetics, and especially the epigenetically mediated inheritance of phenotypic characteristics, on the numerous threatened species receiving some kind of protection. However, given the current and increasing interest in the potential harnessing of various biotechnologies in support of breeding programmes, it is worth exploring what lessons have been learned from decades of research in both human clinical medicine and agriculture. 

The cryopreservation of spermatozoa has been used successfully in agriculture and human medicine for more than five decades, but applications in conservation biology have remained somewhat unpredictable. Semen cryopreservation and successful animal breeding have been undertaken in a range of fishes, mammals, birds, reptiles and amphibians, and this experience has led to the currently increasing interest in semen cryopreservation and the establishment of genetic resource banks (GRBs), often known as biobanks [82,83,84]. The breeding of the Giant panda (*Ailuropoda melanoleuca*) and Black-footed ferret (*Mustela nigripes*) populations provides two, and possibly the only two, examples in which endangered species populations have been successfully boosted through the use of cryopreserved semen, biobanking and artificial insemination [85,86]. Within the GRB context, one important question yet to be resolved, and for which the answers are probably species dependent, concerns the future health and fitness of offspring derived from cryopreserved semen. However, because relatively small numbers of offspring are produced following artificial insemination in threatened birds and mammals, there has been little opportunity to find out whether sperm cryopreservation produces any unwanted effects on offspring. 

Recent advances in functional genomics now offer interesting insights into the ways in which cryopreserved and stored spermatozoa need to be used with great care. Just as some wild mammal populations (for example, gazelles, manatees, and marsupial gliders) are capable of anthropogenically influenced interspecific hybridization, often leading to lowered fertility or even sterility [87,88,89,90], there is an ongoing risk of creating hybrids through the use of poorly organized and inadequately characterized collections of stored gametes. Although it is not yet technically feasible to cryopreserve, recover and use marsupial spermatozoa for artificial insemination, some scientists advocate keeping such frozen collections just in case it ever becomes feasible to undertake techniques such as intracytoplasmic sperm injection (ICSI) [91]. A relevant, and outstanding, example of the care that would be needed involves attempts to conserve the Tasmanian devil (*Sarcophilus harrisii*). This Australian carnivorous marsupial is under considerable threat from inbreeding depression [92], as well as a contagious and fatal form of facial tumour. In 2012, an insurance subpopulation of uninfected animals, derived from two separate founder populations, was established on Maria Island, Tasmania, as one of the conservation measures. Using a combination of microsatellite and single nucleotide polymorphism (SNP) analysis, the researchers showed not only the existence of genetic differences between the two geographically separated founder populations, but an increased level of genetic diversity in the resultant translocated population [93]. Some aspects of the genomic study also pointed to correlations between genes, their respective SNPs, and their functional roles. These included genes involved in embryogenesis, fertilization and the hormonal regulation of reproduction [94] and confirmed earlier findings that inbred Tasmanian devils produce lower litter sizes than expected [92,95]. This conveys a clear and general message about the need for long-term somatic cell, embryo, tissue and gamete repositories to accompany their stored samples with detailed genetic and geographic metadata. 

Data from studies with fishes and amphibians have provided useful and valid information about the survival and fitness of offspring derived from cryopreserved spermatozoa. Recent experimental conservation breeding studies with amphibians (Fowler toads; *Anaxyrus fowleri*) indicated that cryo-derived tadpoles were unusually small and showed poorer post-release survival than those produced without the use of cryopreserved spermatozoa [96,97]. When these authors modelled the likely long-term population impacts of the cryo-derived offspring, they predicted that naturally bred populations would remain stable over a 30-year period, and that populations derived via sperm cryopreservation would also remain stable, but with lower population numbers. A critically important but currently unknown aspect of this work concerns the possibility that the effects of cryopreservation might be heritable across several generations. The authors modelled this possibility as part of their study and found that the population would probably become extinct in approximately 17 years. 

It is not unlikely that the effects of amphibian sperm cryopreservation are heritable, but at present, this is an open question that demands further investigation. This is because the successful use of cryopreserved spermatozoa in breeding programmes for threatened amphibian species, and for establishing biobanks and insurance populations as a hedge against extinction, is widely regarded as an important aspect of amphibian conservation [98,99]. Although there are now several other studies in which phenotypically normal embryos were produced by artificial fertilization with cryopreserved frog and toad spermatozoa, and in which metamorphosis resulted in normal adults (i.e., (*Xenopus laevis* [100] and Golden Bell frog (*Litoria aurea*) [101]), it seems that more research into the heritability question is needed. 

To some extent, these results with amphibian spermatozoa mirror those found by Nusbaumer and colleagues [102] in an experiment in which semen from wild-caught brown trout (*Salmo trutta*) was frozen using a method [103] that produced fertilization rates comparable to those of non-cryopreserved spermatozoa. Cryopreservation did not reduce fertilization rates or affect embryo viability, but did reduce larval growth by about 4%. One possible explanation for the suboptimal success of cryopreserved brown trout spermatozoa is the inhibition of DNA repair within the zygotes [104]. Inhibition of DNA repair mechanisms within oocytes suggested that oocytes defective in some transcripts or proteins involved in repair could block development after fertilization by damaged spermatozoa. As cryopreservation is usually accompanied by the production of reactive oxygen species (ROS) that generate single and double DNA strand breaks [105], efficient cytoplasmic repair mechanisms within the oocyte cytoplasm are necessary to overcome the problem. 

These results confirmed the outcome of a previous experiment undertaken by Kopeika et al. [106]. Embryos were produced from spermatozoa and oocytes collected from a freshwater fish species, the weather loach (*Misgurnus fossilis*). The embryos derived from the cryopreserved spermatozoa were exposed to a DNA-repair inhibitor (3-aminobenzamide—3-AB), and their developmental progress was monitored. Parthenogenically developing embryos were excluded from the analysis, as they do not survive beyond 20 h after fertilization. Sperm cryopreservation significantly reduced embryonic survival during the post-fertilization period. However, to assess the significance of DNA-repair inhibition, the developmental ability of embryos treated with and without, 3-AB was compared after the 20 h post-fertilization stage. The results showed that DNA-repair inhibition significantly reduced the ability of embryos to continue their development (the outcome with 3-AB treated embryos was 84% survival, compared to 91% survival without 3-AB treatment). These results supported the hypothesis that sperm cryopreservation induces DNA damage and interferes with subsequent embryonic development, but also showed that oocytes possess the ability to repair the damage. Significantly, differences in the DNA-repair ability of embryos derived from different females were also observed in this study.

In their 2014 study, Nusbaumer et al. [102] included an interesting extra treatment in their experiment to evaluate the morphology of juvenile brown trout obtained using cryopreserved spermatozoa. Individual embryos derived from the cryopreserved spermatozoa were either exposed to a pathogenic bacterial suspension (*Aeromonas salmonicida*), or only sham exposed as a control treatment. Significantly, exposure to the pathogen increased the incidence of embryo mortality by about fourfold, and those that survived were smaller when they hatched. 

Delayed and abnormal embryonic development was also reported when spermatozoa from the south American freshwater fish species, *Colossoma macropomum*, were cryopreserved using DMSO, dimethylformamide, methanol, ethyl glycol and glycerol as cryoprotectants [107]. All of the cryopreserved spermatozoa had lower methylation levels and exhibited more delays and abnormalities during embryonic development than the control embryos derived with non-frozen spermatozoa. In this experiment the developmental delay started 4 h after fertilization, and glycerol showed the highest incidence of embryonic abnormality with >90% embryonic mortality 11.5 h after fertilization. 

The possibility that fish sperm cryopreservation might lead to inappropriate methylation owing to the use of cryoprotectants was recently discussed in some detail [108], where it was pointed out that the issue is not straightforward. Goldfish sperm DNA methylation was not affected when cryopreserved using methanol, in contrast to the effects on zebrafish spermatozoa, which experienced increased DNA methylation; conversely, DMSO and 1.2 propanediol caused a decrease in DNA methylation. Moreover, these and other authors (e.g., [109]) reminded readers that sperm suspensions from fishes, and indeed from many other species, are not homogeneous, and that sperm subpopulations are not only detectable by a variety of methods, but are likely to be differentially affected by cryopreservation.

## 8. ART and Human Infertility Treatment

The application of ARTs in various human infertility treatments is estimated to have resulted in the birth of over 8 million children worldwide [110,111]. Evidence suggests that a small proportion of the children born following in vitro fertilization (IVF) and embryo transfer suffer from genomic imprinting diseases (Beckwith–Wiedemann, Angelman, Prader–Willi and Silver–Russell syndromes), slightly elevated risk of infant mortality in the first year of life, exhibit signs of large size for gestational age, high birthweight and other problems (for reviews, see [110,112,113]). While some of these problems may be related to the original causes of the infertility, the various technologies used in clinical ART have also been implicated. These include sperm and embryo cryopreservation, embryo culture, and the nature of the culture media used. As many of these technologies are essentially the same as those used in animal studies, it is likely that problems with conservation-relevant animal species would be affected in the same ways but are unlikely to be detectable in the near future as comparatively few such procedures are performed. 

Nevertheless, although it is possible that the sperm cryopreservation process might skew the success of assisted reproduction methods in general, it is encouraging that the extensive biomedical and research use of cryopreserved, and banked, mouse spermatozoa (where ICSI is widely preferred as the fertilization method) do not appear to have produced the range of problems seen with fishes and amphibians [114,115].

## 9. Stem Cells, Conservation and Ethics

The cryopreservation of fish and amphibian oocytes presents a difficult problem, because these are large cells filled with aqueous cytoplasm, typically enclosed within impermeable and tough outer coatings that inhibit, or prevent, the penetration of cryoprotectants. An alternative approach that works for fishes involves the cryopreservation and transplantation of primordial germ cells (PGCs) into sterilized surrogate recipient individuals at different life stages [116,117,118,119]. PGCs are the precursors of gonocytes, the stem cells that migrate to, and colonize, the genital ridges of early embryos, where they populate the testes and ovaries, and are eventually responsible for producing spermatozoa and oocytes. When the PGCs have originally been obtained from a donor species that differs from the surrogate recipient, the spermatozoa and oocytes produced thereafter are derived from the PGC donor. Thus, PGCs are widely considered to represent a practical solution for the cryobanking of fish species, as they conserve both the paternal and maternal genomes. Fewer authors have reported undertaking successful PGC transfers in amphibians. *Rana pipiens* PGCs, dissected from tadpole genital ridges [120], were successfully transferred, into enucleated eggs, obtaining normal development in about 40% of the transfers. and demonstrating the totipotency of the germ cells. A later study [121] also demonstrated the principle of amphibian germ cells totipotency using salamanders. These authors transferred diploid germ cell nuclei into irradiated eggs, and obtained not only an adult male and female, but also some normal offspring. Despite the current catastrophic extinction rates being experienced globally by amphibian populations, there has been very little interest in developing PGC transplantation methods in aid of amphibian conservation. This may simply have been a matter of scientific priorities, with aquaculture research providing the stronger motivation. However, a recent review [122] nevertheless expressed the view that these approaches might still be worthwhile exploring. 

In practice, there are several variations of the techniques involved in gonocyte transplantation, some involving transfer of PGCs into recipient blastulae, while others involve transplantation of PGCs, spermatogonia or oogonia into hatchlings, or spermatogonial transfer into adults [123]. Once the cell suspensions are microinjected into the intraperitoneal cavity, the PGCs migrate to the genital ridges where they develop and produce either oocyte or spermatozoa. Rivers et al. [117] pointed out that the PGC transplantation method appears to be more suitable for conservation biobanking than other conventional methods of gamete storage.

Spermatogonia [124], as well as PGCs, can be cryopreserved easily and, when transplanted, they have the potential to produce millions of gametes and offspring throughout the life of the recipient. Spermatogonial transplantation is widely seen as a potentially successful method for treating testicular cancer in prepubertal and adolescent humans, who have to undergo chemotherapy or radiotherapy. Some approaches require that the patient’s spermatogonia are recovered before treatment and subsequently used to recolonize the testes. Other approaches require the spermatogonial recovery to include various intermediate steps, including the temporary culture of cells within the bodies of immunodeficient animals such as nude mice. All of these options have been summarized in a recent review [125]. With a view to conservation, some of the techniques have been applied to non-human mammalian [126,127,128,129,130] and avian species [131,132], as well as fishes. 

Subjecting one group of (non-threatened) animals (typically immunodeficient and sterilized mice or their equivalent) to highly invasive surgery, such as inserting testicular cell aggregates or cell cultures beneath the kidney capsule, is ethically dubious if the objective is to protect a different group of animals, even though they may be considered to be at heightened risk of extinction (see [129,133,134,135] for examples of these techniques). Careful ethical justification is needed when proposing such steps for species conservation, including the valid statistically based reassurance that the offspring are not adversely affected because of the varied technologies required. 

The explosion of interest in the exploitation of various stem cell technologies for the production of spermatozoa and oocytes in vivo, in order ultimately to produce embryos, has not only raised new technical possibilities for species conservation, but has also generated considerable ethical debate. The discovery that pluripotent stem cells (iPSCs) can be generated from mouse embryonic cells and adult fibroblasts [136] has been followed by the generation of iPSCs in multiple species [137,138]. This has led many biotechnologists to hypothesize that it will eventually be possible to produce functional gametes, and therefore also embryos, from rare and endangered species without the need to involve other animals such as the immunocompromised and ovariectomized mouse [134,137,139]. At present, the success rate would likely be rather low, and given the number of technical steps needed, the health and survival of any offspring might also be low, especially as the detailed reproductive biology of most threatened species is poorly documented. 

Although accessing suitable tissue and cell samples from rare and endangered species for the purpose of creating stem cells is not a problem with captive and domestic animals, it would undoubtedly present problems with free-roaming wild species. Deriving stem cells from skin samples might provide a more practical solution to this problem [140], as wild species, including marine mammals [141], are often biopsied remotely for genetic studies, using puncture darts and without the use of anaesthetics. Although this approach still requires further development, stem cells successfully derived from porcine and murine skin have been shown to possess the capacity to form oocyte-like and PGC-like cells [142,143], as well as other non-germline cell types. 

## 10. Conclusions

To survive on earth, all organisms have developed mechanisms that permit them to cope with the environment in which they find themselves. Newborns cannot usually choose the locations in which they would prefer to live, but some species have evolved mechanisms that allow for some degree of parental choice in this matter. For example, many reptiles have evolved temperature-dependent sex determination, whereby the females are able to predict the expected temperature that eggs will experience in the nest, thus skewing the offspring sex ratio [144]. In some reptiles, this mechanism operates alongside a genetically based sex determination system, and recent evidence has shown that the two systems can even be operated alternately and independently [145]. This is an astonishing example of adaptive developmental plasticity but, as described above, it is only one among many different solutions to the problem of forecasting the immediate environmental future and responding appropriately. 

In this sense, environmental change includes not only the warming climate and its consequences, but even some of the well-meant mitigation measures that are under human control. As discussed above, the captive breeding programmes used widely by zoos, wildlife parks and aquariums as conservation measures, or even simply providing food to wild animals such as bottle nose dolphins [146], appear in some cases to reduce the fitness of the animals that are being protected. Should zoos and nature reserves respond to such findings by reducing welfare and introducing competitors, predators and parasites, thus creating more adverse conditions? This would undoubtedly be a counter-intuitive, unpopular, unethical, and even illegal measure, but it would presumably lead to the selection of individuals that are (a) adapted to life in a captive environment, but (b) also able to survive if relocated to a more natural but possibly less comfortable environment.

On a related topic, it is evident from the preceding discussion that while the threatened populations may benefit from technologies such as genetic supplementation with cryopreserved and stored spermatozoa, there may be an inherent risk to the health and welfare of offspring. Whether the introduction of more advanced breeding technologies, especially those that will aim to regenerate the rarest species using stem cell reprogramming and artificial gametes, would exacerbate the risk is largely unknown at present given that numerically large datasets are needed before minor effects can be detected. Detailed analyses of developmental problems in laboratory and agricultural mammals derived from somatic cell nuclear transfer, and more recently the use of iPSCs, have revealed multiple problems (summarized in [147]) that frequently lead to an abnormally high incidence of perinatal mortality, inadequate placental function and abnormal offspring [148].

Whether and when to use such advanced technologies in conservation, where the objective is to produce viable and healthy offspring with the capacity to survive and live a healthy life, raises many difficult and controversial decisions. For example, is it acceptable to deliberately produce animals that are highly likely to be unhealthy and abnormal? As some of the species of interest require several years to reach breeding age, where might such unhealthy, and possibly disabled, animals be kept? In the exceptional circumstance that one or two members of an extinct species, such as the woolly mammoth, is successfully regenerated using advanced technologies [149], would the absence of appropriate maternal care and social interaction be a problem? Detailed ethical discussions around this topic are outside the scope of this article, and the reader is referred to other articles for further information [150,151,152,153].

## Figures and Tables

**Figure 1 biomolecules-12-01297-f001:**
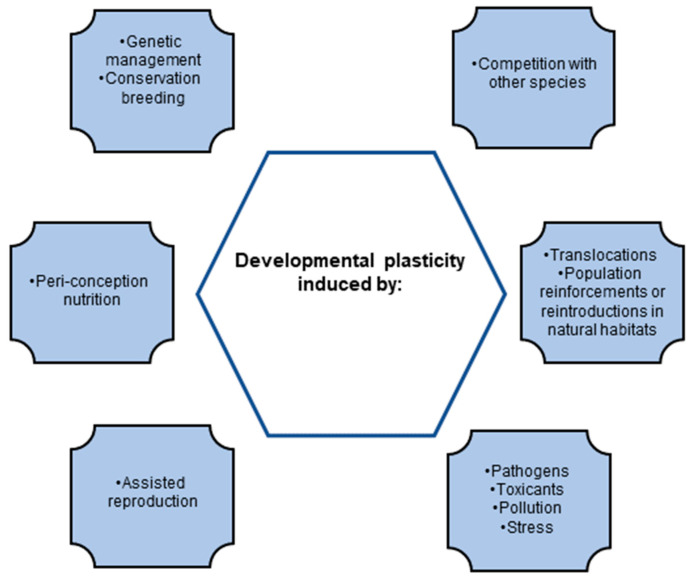
Factors that induce developmental plasticity.

**Figure 2 biomolecules-12-01297-f002:**
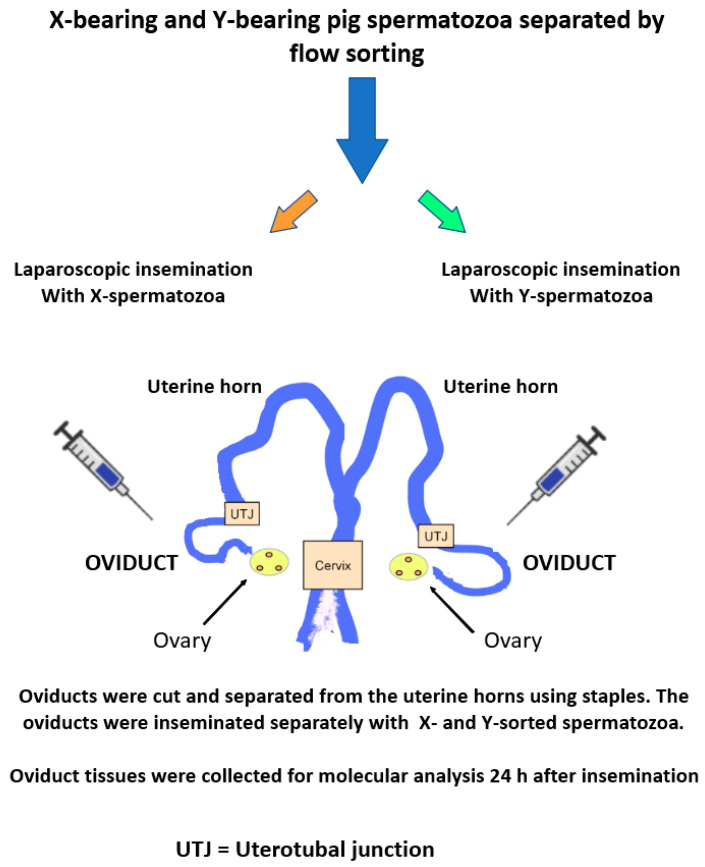
Schematic diagram showing the method used to establish that the oviducts can distinguish between X- and Y-chromosome-bearing spermatozoa.

## Data Availability

Not applicable.

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
