# Peer review of "Conservation Biology and Reproduction in a Time of Developmental Plasticity"

_biomolecules, 2022, doi:10.3390/biom12091297_

Round 1
Reviewer 1 Report
General comments
Very interesting and well written manuscript. Very timely indeed
Howevere there were no line numbers making it more difficult to refer to parts in the text
At some points, some editing may be required to facilitate reading.
I wonder if the authors care to comment in their manuscript on the ethics related to the recently assigned scientific programmes to de-extinct the woolly mammoth https://reviverestore.org/projects/woolly-mammoth/ and the Tasmanian tiger? https://www.bbc.com/news/world-australia-62568427
Specific comments
Abstract
I find the following paragraphs difficult to understand. It is important to clarify it because readers will decide based on the abstract whether they will read it or not
“Importantly, conservation breeding programmes, used widely by zoological parks and aquariums, may appear in some cases to lower fitness, by reducing animals’ abilities to cope when confronted by the ‘wild side’ of their natural habitats.”
Since lower can be used as adjective and as verb, it may be preferable to replace it by decrease. I read “lower” at first as an adjective and found it difficult to understand
“Would more adverse conditions lead to the selection of individuals that are adapted to life in a captive environment, while being unable able to thrive if relocated to a more natural, but possibly less comfortable, environment?”
I do not understand the meaning of this sentence : I understand that you should impose more adverse conditions to make them hardier, and even then they do not survive in the wild? This makes no sense to me.
6. Wildlife conservation, captive breeding and mammalian sex ratios
Achieving these outcomes at the population level can only occur if individuals are able TO ? detect and respond to environmental cues experienced very early in the reproductive process, even before conception in the case of species with internal fertilization
Reviewer 2 Report
This interesting manuscript make the point that developmental plasticity can potentially negatively impact conservation breeding programs by lowering fitness of an individual to cope with release into wild habitats. Several excellent examples are cited demonstrating negative effects of ART, climate change, gamete cryopreservation etc. However, the authors argument for developmental plasticity is then conflated with zoo animals being poor candidates for reintroduction, which can (and possibly more likely) also be attributed to a lack of resilience rather than epigenetic effects. Although species that are environmentally driven to reproduce or produced using ARTs may face challenges in introduction programs, others produced through natural breeding, such as the scimitar horned oryx, for example, seem to do well, though comparatively low numbers and length of time since release preclude data analysis to confirm or deny otherwise.
Captive raised species that are introduced into wild habitats are usually managed differently than zoo animals, with different diet, enclosures and a reduction in human interaction etc. There are frequently also soft release programs that form part of the release to acclimate the individual to the new habitat, including predator avoidance training. Can the authors comment on whether they think developmental plasticity plays a role in reducing fitness of introduced species in spite of reintroduction management systems that lend themselves to increased resilience?
Zoo animals are rarely introduced into the wild, and their role is predominantly that of research, education and entertainment, so creating a more adverse environment to increase resilience would not be necessary.
Other than the argument that zoo animals would do poorly in reintroductions, with which I do not entirely agree, this is a fascinating and timely discussion on the effects of epigenetics on species biology, highlighting hazards, known and as yet, unknown, of ARTs, STEM cell technology etc. that may lead to inherent risks in health and welfare of individuals.
A final comment, the concluding sentence regarding ethical decisions is rather dour indicating people embrace the challenges without hesitation (suggesting imprudence), while other adhere sternly to the precautionary principle. Might there be some middle ground where ethical decisions and challenges involving cutting edge ARTs are viewed practically but optimistically, and with the recognition that there will always be limitations in new technologies but that over time these will be mitigated and eliminated through continued scientific advances.
Reviewer 3 Report
The authors proposed this review article to ask whether, and how, principles in conservation biology may need to be revisited in the light of new knowledge about the power of epigenetics to alter developmental pathways. This article was an interesting and obviously a nice piece of review article. The reference articles used to explain their views on the current status of the research area are adequate and well described. This review was well written and could be acceptable for publication.
